# An eye for talent: The recruiters' role in the Australian Football talent pathway

**Paul Larkin** [1,2]*, **Daryl Marchant**[1], **Amy Syder**[3], **Damian Farrow**[1]

**1** Institute for Health and Sport, Victoria University, Melbourne, Australia, **2** Maribyrnong Sports Academy, Melbourne, Australia, **3** Institute of Psychiatry, Psychology & Neuroscience, Kings College London, London, United Kingdom

* paul.larkin@vu.edu.au

**Data Availability Statement:** All relevant data are within the paper and its Supporting Information files.

**Funding:** PL DF DM received funding from the Australian Football Research Board for this study.

## Abstract

Talent identification and recruitment (TIR) in elite sport is a complex process with performance and career longevity implications for athletes, sports organisations, and scouts (hereafter referred to as recruiters). Although there is an established body of published research on TIR the critical roles that recruiters perform has only recently gained attention from researchers. In this study, we report on the practices that 12 full time Australian Football recruiters use to inform their TIR decisions. Inductive qualitative semi-structured interviews were conducted using open-ended questions. Thematic analysis resulted in the identification of four primary themes (the recruiter, processes and practices, assessment and selection). The results and discussion provide insights and an occupational 'road map' into the important role recruiters perform in sporting organisations.

## Introduction

The process of talent identification and recruitment (TIR) is about making informed decisions to select the most promising athletes, with best potential to excel as an elite athlete [1–4]. TIR is also an integral part of the talent pathway and results in substantially divergent paths for athletes (e.g., early identification into an elite program; late identification from community based programs). Anecdotally, recruitment decisions are retrospectively scrutinised *ad nauseam* by sports fans, the media and within sports organisations. Although there is an established body of published research on TIR [see 5, 6] the central role of recruiters and how they perform their roles have largely been ignored.

Australian Football League (AFL) clubs recruit players through a National Draft, and a Rookie Draft with approximately 120 players selected annually. The talent identification and selection process are important because selection and non-selection can have significant impacts on the careers and life directions of the athletes' [7], and the success and direction of the AFL clubs. Retrospective assessments of draft selection outcome are essentially examining predictive validity because initial draft rankings are being assessed as predictors of specific criterion outcomes (e.g., sport-specific performance measures, career longevity or other relevant variables). In an attempt to examine specific predictive indices researchers have assessed numerous variables (i.e., physical; anthropometric; technical and psychological) that may differentiate between selected *vs*. non-selected athletes or successful *vs*. unsuccessful athletes [8–13]. For example, from a physical and

The funders had no role in study design, data collection and analysis, decision to publish, or preparation of the manuscript.

**Competing interests:** The authors have declared that no competing interests exist.

anthropometric perspective researchers have indicated players selected in the national draft are taller, possess higher levels of running endurance (i.e., 20m multistage fitness test) and are faster (i.e., 20m sprint time) compared to non-selected players [10]. Also, in-game performance variables have been shown to differentiate drafted and undrafted players [14]. Furthermore, tactical performances, via video-based decision-making tasks can discriminate between talent-identified and non-talent identified players [15].

Talent identification is complex, with multidimensional performance factors, including; physical, physiological, technical, tactical, psychological and sociological influences [16–20]. While previous investigations in Australian Football [8, 10, 12, 15, 21, 22] and other invasion sports [23–26] have assessed factors that contribute to the talent identification process, few researchers have purposively engaged with key stakeholders, such as recruitment staff, responsible for identifying and selecting talented athletes [2, 27]. Larkin and Reeves [3] recently called for a shift in perspective when conducting talent identification research to understand the processes, observations, and perceptions of recruiters when making talent identification decisions. By understanding the rationale and decision making of recruiters, invested individuals can potentially be more informed (i.e., recruitment staff, coaches, athletes, and parents).

To understand talent identification and recruitment decisions, researchers have employed qualitative designs such as interviewing experienced soccer coaches. Christensen [28] conducted in-depth interviews with eight Danish national team coaches and found that coaches valued game intelligence (i.e., ability to read and predict game-play), and soccer-specific physical and technical skills as the most important variables when assessing talent. Coaches also considered personal qualities, including; character, attitude, drive to succeed, and willingness to learn as important. Larkin and O'Connor [2] supported these findings by identifying a hierarchy of attributes perceived as important by coaches when identifying youth soccer players; technical (i.e., first touch; striking the ball; one-versus-one ability; technical ability under pressure), tactical (i.e., decision-making ability) and psychological attributes (i.e., coachability, positive attitude). Lund and Söderström [29] focussed on how 15 Swedish soccer coaches made talent identification decisions with the key themes reported being; previous experience, current elite player's qualities, and the values and belief system of the club. From an Australian football perspective, MacMahon and colleagues [27] developed a preliminary model for understanding factors that influence recruitment in the AFL. The model identified four key factors, including (i) recruiter background; (ii) recruiter attributes; (iii) recruiter understanding of team needs and (iv) recruiter–coach relationship. The findings highlighted the athlete recruitment decision-making process is based on intuition and deliberation, and influenced by the recruiters relationship with the head coach. While MacMahon and colleagues [27] identify the decision-making process of recruiters, there is still limited understanding of how recruiters make talent decisions, in particular what information they value when discriminating between athletes to recruit.

While the current talent identification and selection literature highlights the decision-making processes [27, 29] and the associated athlete attributes they consider important to consider in this process [2, 28], there is still limited understanding of the recruiters role within the talent identification process, especially from their perspective. Therefore, in the current study, we describe AFL recruiters role within the talent identification process in Australian Football, and what information they use to make talent identification and recruitment decisions.

## Method

### Design and participants

Interpretive phenomenology (IP) was adopted to examine the work or role experience of full-time sport recruiters. Phenomenology is generally used to explore the central meaning of a

shared experience [30]. We chose interpretive phenomenology because it allows greater scope for research input compared to descriptive phenomenology. Purposive sampling was used to enable the 'recruitment' of participants with deep and sustained experiences as a recruiter. Twelve full-time male recruiters (i.e., scouts) from 12 of the 18 AFL professional clubs participated in this study. Eleven of the twelve participants were National Recruiting Managers (with the other one being the assistant national recruiting manager) with extensive experience in Australian Football (AF) talent identification (average years in AF talent identification = 15.1 years; SD = 8.5). Overall, all participants had extensive experience, ranging between 10–47 years, working within Australian Football clubs (i.e., coaching; playing; sports analysis). Victoria University Human Research Ethics Committee gave ethical approval for the study. the approval number is HRE16-193, with informed consent provided by participants.

## Procedures

Inductive semi-structured interviews were the method of inquiry. Open-ended questions were developed from previous talent identification literature [2, 27] and were used to promote a broad discussion relating to the participant's talent identification knowledge (e.g., How do you define talent in Australian Football? What is your talent identification philosophy?) and practices (e.g., Tell me about how you assess and make judgements on a player's ability?). The interview schedule had nine main sections (e.g., recruiting and talent selection background; deconstruction of a recruiter; recruiting practice; talent assessment; athlete attributes; talent selection decision-making; talent development pathways; athlete development; recruiter analysis) which included open-ended questions (e.g., Talent is a multi-dimensional area therefore how relevant is technical ability when identifying a potential athlete?) and additional probing questions to enable participants to explain how their specific knowledge and practices assisted their decision-making (e.g., Why is this attribute important when identifying players?; Can you provide an example from a game context?; how do you identify the attribute/characteristic in an athlete?). The interview schedule is available on request to the lead author. The lead author conducted all interviews which were recorded and ranged in length from 60–75 minutes. Interviews were transcribed verbatim comprising 269 pages of text.

We have considered each of the four pillars of trustworthiness proposed by Guba [31] namely, credibility, transferability, confirmability and dependability. To establish credibility we used prolonged engagement in the field, internal peer debriefing and member checking. Engagement in the field translates to researchers spending time in the field of inquiry [32]. As authors, we have engaged intensively with AFL clubs, in particular, the second and third authors being professionally immersed into AFL clubs for a combined total of 32 years. We contend that this sustained involvement with AFL recruiters, coaches and players has been central to establishing a deep understanding of the participant's culture, context, and core issues in recruiting, taking a longitudinal perspective and building rapport within the AFL recruiting fraternity. Krifting [33] observed that rapport with informants translates into the sharing of more sensitive information. As researchers and authors, our sustained involvement in AF was beneficial in accessing participants, eliciting quality data, and conducting the data analysis. We used peer debriefing and reflexive conversations as an internal loop to discuss and modify all aspects of the study. Member checking involved all participants receiving copies of their transcripts, and providing feedback on the accuracy of the data.

To establish thick transferability we used purposive sampling. Thick descriptions, as described by Anney [34] and Li [35], involves providing readers with detailed descriptions of the study methodology, context, design and procedures to enable other researchers to replicate the study in similar or other contexts. We attempted to provide sufficient details to enable

other researchers to replicate aspects of the study, but also invite communications from researchers to request further detail if required. Purposive sampling was used to recruit AFL recruiting managers as a discrete group of informants because of their likely capacity to provide in-depth information on all aspects of the talent pathway in professional AFL football.

To establish dependability, we have maintained documents typically required for an audit trail (i.e., internal and external communications, informed consent, interview guide, raw data interviews, transcriptions, a procedural spreadsheet and field notes). We also used stepwise replication and peer examination, whereby each author independently analysed the data and compared the results to determine consistencies and inconsistencies, such as the thematic structure, coding and representative quotes selection. Finally, we have attempted to establish confirmability by cross-referencing our results with similar studies (see discussion), and also question the objectivity and potential influence of our own experiences, knowledge, and biases on the resultant data and conclusions. Nevertheless, confirmability is somewhat dependent on conclusions from other researchers based on future TID research. Despite our efforts to establish trustworthiness, we acknowledge that the themes and interpretations presented can never be entirely inseparable from our frames of reference. Similarly, the results are trustworthy as a series of data slices constrained by the topics explored, time and circumstance.

Following data checking, the first and second author separately read the transcripts until they were familiar with the content and began by first coding the raw data (i.e., meaningful quotes). This process led to an initial identification of 18 themes considered important to understanding the processes associated with talent identification and recruitment in AF. We then collectively discussed and operationally defined each preliminary theme with minor alterations made to reconcile differences in the original coding themes constructed by the first two authors.

## Results

Four first-order themes (FOTs) emerged from the thematic analysis; (1) The Recruiter; 2) Processes and Practice; 3) Assessment and (4) Selection (see Fig 1). We identified 25 second order themes (SOTs) and 179 third order themes (TOTs). We have structured the results to provide a brief introduction to the four FOTs followed by an overall summary description and indicative quotes for each of the 25 SOTs. We have also provided an appendix listing the 179 TOTs. The quotes provided are indicative of one or more of the TOTs; however, due to space restrictions, the number of TOTs represented is relatively limited.

### The recruiter

The first order theme (FOT), the recruiter comprised seven second order themes (SOT) and 46 third order themes (TOT) relating to the recruiter. The SOTs were, background (7 TOTs), personal characteristics (8 TOTs), recruiting philosophy (6 TOTs), knowledge and experience (8 TOTs), skill set (5 TOTs), challenges (6 TOTs) and professional development (6 TOTs).

The SOT theme *background* comprised TOTs such as a love of football, playing experience and initial entry as either work experience, being part-time or a volunteer. For example, "You need to have a real passion to go and watch football" and "It's good not to have lived in the bubble that is the football industry your whole career." In terms of background, there was considerable diversity in the background of recruiters; "I'd done a major in Human Resources. . . . and then I got into property and was doing another degree where I did a lot of predictive mathematics." The SOT, *personal characteristics* included themes such as strong work ethic, attention to detail, self-belief in their decision, being open-minded and having a growth mindset. Furthermore, sound communication skills, being open to others opinion and building

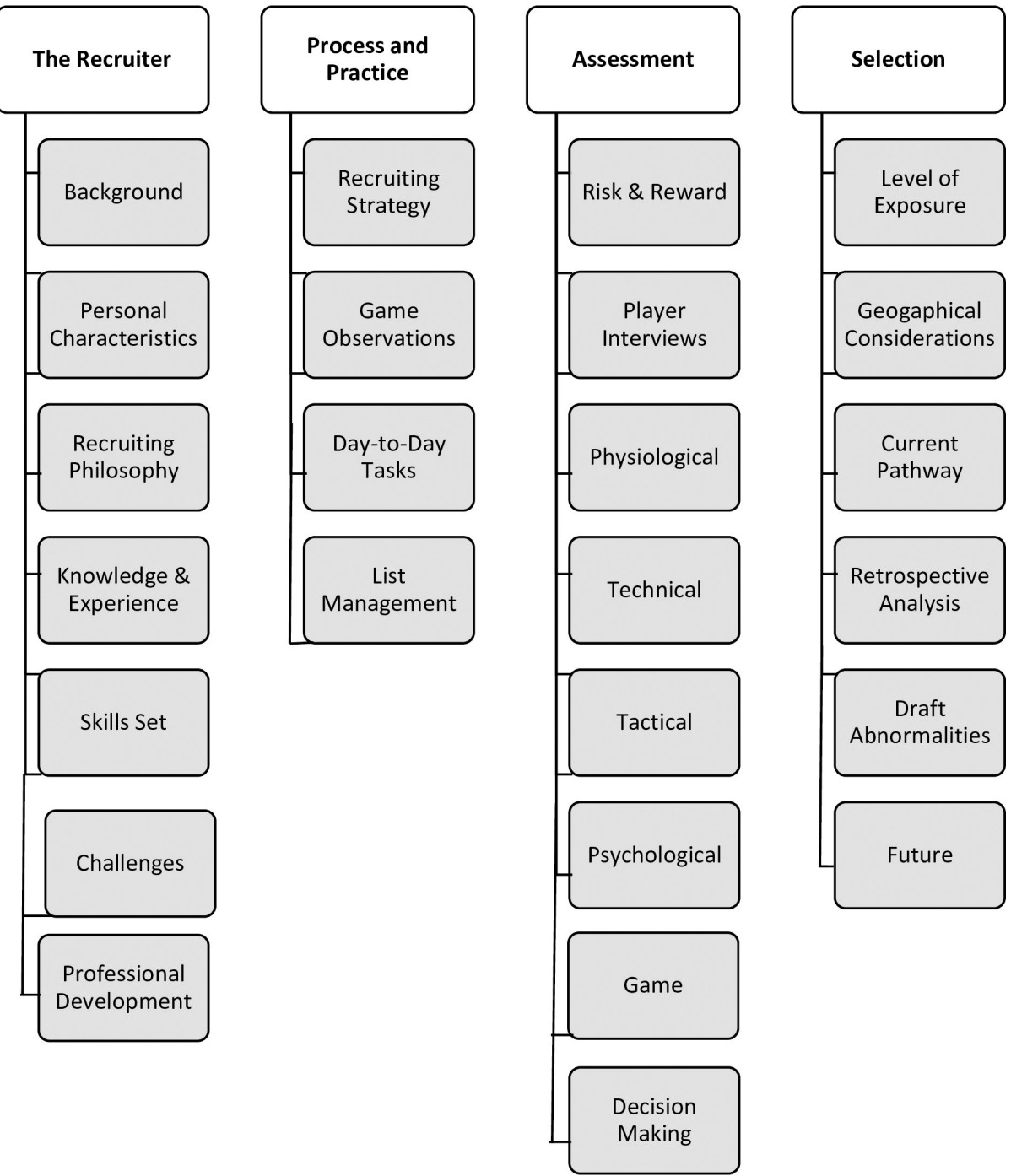

**Fig 1. Schematic of first order themes (white boxes) and second order themes (grey boxes).**

strong working relationships were frequently discussed by recruiters. For example, "not being stubborn about one's opinions and being open-minded to what the future will hold" and "We're constantly trying thinking of ways how we get better." The SOT, *recruiting philosophy* demonstrates the underlying approach to recruiting and includes; a clear understanding of talent definitions, how to develop talent, perceptions of vital or required attributes and matching

personal philosophy with a broader club philosophy. For example, "We have a list manage-ment model at the club. It's up to the recruiting staff to find players that fit the model." Recruit-ers had slightly similar perceptions of what constitutes talent, with ideologies based on the skills and consistency of performance of the skill. For example, "talent is the footy skill, the game sense, the ability to play the game, probably a fair bit of decision making", in conjunction with "a skill that an individual can repeatedly perform that the majority cannot and the ability to execute a series of skills on a consistent basis."

The SOT, *knowledge and experience* were largely about accumulated skills over time and developing specific and nuanced understandings. Underlying TOTs commonly discussed was the ability to make reliable comparisons, learning from mistakes and understanding young people. For example; "But you've got to make those mistakes when. . .it won't cost the club 300 grand" and "Teachers probably do pretty well in recruiting because they know young people." A key aspect of developing these skills was experience for example one recruiter said, "experi-ence is nearly the most important thing because you need to know a whole pool (of players) and you need to have seen or know a lot of the kids or known a lot of young people to know what they are like. One recruiter suggested that an experienced and capable recruiter "does not jump to a hasty conclusion, but takes in all the information and waits to make a very consid-ered decision on who will be the best player for the longest period." It is this experience within the talent identification process that shapes their philosophy of talent, and specifically what they are looking for in potential players.

The SOT, *skill set* refers to specific job requirements over time such as the ability to predict and identify potential talent. Possessing strong analytical and observational skills are impor-tant and for some recruiters being able to manage a team and using information judiciously to inform/change decisions. For example, "I see something in a game that makes me think they might be able to transition into AFL" and "I think my background from a business point of view in managing the operation and from a strategic point of view is important."

The SOT, *challenges* comprises of numerous TOTs including; work-life balance, due to the amount of managing travel, and reduced family time. Also, there is pressure associated with the decisions made having lasting ramifications for the organisation and recruiters themselves. For example, "whether the player turns out to be any good or not that's always going to come back to you" and "You probably actually do a better job if you've got balance." As with most vocations, the SOT *professional development* is about the scope for linking to a broader net-work of industry colleagues, continual skill development and keeping abreast of international trends. For example, "I've caught up with other sport, you know NBA, NFL and local teams just to try and pick their brains" and "I've done a lot of courses that have helped me."

## Practices and processes

The FOT processes and practices comprise 4 SOTs and 31 TOTs relating to the daily practices of recruiting work and processes used to organise the work; recruitment strategy (7 TOTs), game observations (8 TOTs), day-to-day tasks (8 TOTs) and list management (8 TOTs).

The SOT *recruiting strategy* is what guides day-to-day practice. The related TOTs including; the current talent pool, the quality of information networks, systems for constant reappraisal and comparisons, time efficiency and using experts where relevant. For example, "It's probably rare to get a combination that's perfect, so you've got to put a value on what you prefer" and "It's all taken into account, cost and benefit risk-reward, every decision we make is what's going to be best for our team and club." Because recruiters spend so much time watching ath-letes the SOT *game observations* was reflective of considerable investment in time and practice. With live and digital footage of gameplay commencing for players at age 16 and intensifying

through to under 18 a strong focus is placed on, the national championships, game day reporting processes, scope and organisation of the internal recruiting network and repeated observations to improve reliability. For example, "The Nationals are really important because it's the best against the best at their own age" or "We might have already seen them play 30 times." The process of tracking a player involves watching games and "then going back and revisit the vision from that game (recorded video footage of the game)." This process of watching live games is an important step, as many recruiters indicated they "watched every player we drafted live at least 20 times between us (recruiting department) plus another 20 to 50 on vision (a video recording of a game)." The SOT, *day-to-day tasks* focus on the minutiae of recruiting practices. Typical TOTs were the recruitment and management of staff, data collection and cross-referencing, following up on discrepancies, referencing checking and internal mock drafts. This type of daily work is the fine-tuning and calibration of the system and practices. For example, "The hardest thing is not seeing [all] the games. . . other clubs have got six staff members their coverage is massive" and "You might have hundreds of names. . .and then you start to funnel that down." The SOT, *list management* refers to the numerous considerations, rules, and opportunities to balance the overall club player profile. Typical TOTs are; long term planning, free agency, second chance players, the priority of coaches, rule changes, the actual number and order of ensuing draft selections. For example, "How your coach wants your team to play, what he needs and his game style" and "When you get to the back end of the draft or the rookies that's when you may slant towards a need."

## Assessment

The FOT assessment comprises 8 SOTs and 60 third order themes (TOTs) relating to assessing a myriad of attributes of prospective players; risk & reward (9 TOTs), player interviews (7 TOTs), physiological (8 TOTs), technical (7 TOTs), tactical (3 TOTs), psychological (11 TOTs), game (9 TOTs) and decision-making (6 TOTs). One recruiter succinctly summed up the importance of assessment; "we tend to assess each player individually similar to a balance sheet, their assets and their liabilities and can the liabilities be improved and how strong is the asset."

The SOT *risk and reward* relates to the perceived assets and liabilities of a player when selecting. The common industry perception is that players touted as high draft picks are 'safe' options, with later picks perceived to be 'riskier' options. Recruiters talk about mitigating risks by assessing character, medical and injury history, social media. For example, "there is more pressure on the first picks, and they tend to be a lower risk anyway because there are less flaws in their game" and "we shy away from someone who is going to come in and cause a lot of trouble for the coaches and off-field staff." Regarding athlete selection, recruiters highlighted the importance of the *player interview*. This SOT demonstrated that often 2–3 interviews were conducted with each potential or shortlisted player to develop an understanding of the player, their background and family. For example, "the family background, the history with parents and the family dynamic can sometimes impact." Coupled with interviews are also the standard practice of "references from coaches and player managers." Some recruiters also talked about contacting school teachers to provide references.

The SOT, *physiological* are focussed on physical attributes that influence recruitment decision-making. Typical TOTs were the anthropometric, athleticism, genetics, endurance, medical history and scope to improve. One recruiter remarked, "for example [player name removed], his time trials and stuff, it's good, but it's not great. But you look at his GPS and game stuff and it's through the roof, it's elite" and "I think what initially maybe draws you to, or they come to your attention, is they've normally got one or two athletic or skill qualities that

are quite good." Further to this point, the SOT *technical* highlight the importance of game-based technical skills on the recruitment decision-making process. Typical TOTs are; ball winning, kicking, marking, and the assessments that are less relevant or valued depending on the recruiter's preferences. For example, "Yeah we place enormous emphasis on kicking, but that doesn't mean they have to be a great kick right now." Conversely, "kicking is one that stands out a lot more because it's a little bit harder to master and if you're a poor kicker there's not much you can do about it sometimes, so kicking is critical." Some recruiters also questioned the validity of assessment methods, for example, "the kicking test at the AFL combine would have no influence at all on our decision making". The SOT *tactical* attributes also provided evidence of the type of skills recruiters looks for when identifying a player. TOTs included game sense and being able to get involved at the right time of the game, which highlighted the importance of game understanding for a player. For example, "have an ability to just, read the cues . . . know that the ball is going to be in a certain spot at a certain time and they're there when it gets there" and "More than ever you need to know and understand set-ups and systems and multiple ones because the game changes all the time."

The SOT, *game* refers to aspects of game performance that contribute to the recruiting decision-making process. Typical TOTs are; consistency of performance, versatility, ability to influence the game, and team orientation. For example, "A player who can do everything. . .the actual skill level, footy-smart, and the athletic ability, ground and air and you know versatile." and "he mightn't be best on the ground,. . . but those 10 possessions might influence the game." The SOT *decision-making* relates to both on-ball and off-ball game decision-making, with this process contributing to the effective execution of skills. TOTs included decision-making under pressure, execution of skills, 'ball in hand' decisions and 'off ball' decisions. For example, "place a bit more emphasis on the decisions in critical moments . . . does he do the same under pressure as he does when he's not under pressure?" and "You can be technically the best kick or hand-baller or executor of any skill, but if you make the wrong decision then it's totally flawed." Recruiters also discussed the importance of decision making in running patterns and game sense, "their game sense in the game, knowing where to run. That is a big one for us because that is a pretty hard thing to coach" Finally, the SOT *psychological* highlighted how recruiters liked players who have a strong work ethic, determination, drive and attitude to compete at AFL level. Without exception, recruiters indicated the psychological assessment is a vital piece as "there are really talented players they can do most of the skill facets really well, but mentally they are not going to handle the pressure of elite competition there is no point even bringing them in" They also highlighted the possible "red flags" that may influence the decision-making process. Typical TOTs included resilience, work ethic, attitude, determination, character and competitiveness. For example, "So players that are competitive, they're driven, love the game, love to train, love to work. When the heat is on they stand up and they've got good values, good people, fit in well in a team environment" and "we defer it to the experts to say there's a red flag here or there's nothing to worry about, and if there is a red flag let's interpret it." The psychological assessment can be a make or break assessment for some recruiters, "I will never pick a kid if I didn't think he was mentally up to it, even if he was an enormous talent."

## Selection

The FOT selection comprises 6 SOTs and 42 third order themes (TOTs) relating to the intricacies that inform the actual selection decisions; level of exposure (4 TOTs), geographical considerations (6 TOTs), current pathway (9 TOTs), retrospective analysis (9 TOTs), draft abnormalities (7 TOTs) and future (7 TOTs).

The theme *level of exposure* (SOT) comprised TOTs such as wanting to see players competing at the highest level and the concept of data saturation. AFL recruiters teams are well staffed have vast data collection networks to ensure extensive data is available on all players of interest within the pathway. A strong emphasis is placed on the player's performance at the U18 National Championships. For example, "strong emphasis on the under 18 championships because 80% of the drafted players are from there."

The SOT, *geographical considerations* highlighted the practice of virtually weighting the relative advantages and disadvantages of players living and competing in remote locations for instance. There is a perception among many recruiters that country recruits often have more upside (i.e., the potential for improvement) due to the limited exposure to a professional training environment compared to city recruits. For example, "county kids train together once a week, city kids 2 to 3 times a week. Country kids might be driving 2 to 3 hours just to training let alone games" and "If the player lives in a remote part of the Victorian country... Well, you'd have to think that if he's in an AFL programme every day, he's got a lot more improvement." The SOT, *geographical considerations* highlight the influence of players' often being required to move significant distances from their family once recruited and how their perceived maturity and readiness to live more independently is considered. Therefore, if two players are of a similar skill level, recruiters are often likely to pick the local player to reduce the later potential for 'flight risk' associated with some players. For example, "we would rather pick kids who might not have as much talent, but we know they would be in it for the long haul and have come for the right reasons" and "they might be from interstate, and you think oh they're not going to hang around here."

The SOT, *retrospective analysis* refers to the specific process of making predictions based on comparing them directly to how a current elite level player was performing at the same age. Recruiters acknowledge this is not an exact science, but with the increased access to resources and data, they believe they are getting better at identifying the best potential players. For example, "we go back and we match them against each other (i.e., current and former player) and this is why it's so important for us to have replicated the data over the years" and "We're not perfect at it (predicting players), but I think we're getting better at it and certainly we should be getting better with the resources that are thrown at it." The SOT, *draft abnormalities* highlight potential factors, whereby a player initially overlooked in the draft, or a low draft ranked player subsequently develops into a 'champion' or 'superstar'. Factors identified included recruiters over or undervaluing certain player traits, or a player may be unlucky as only a certain number can be selected, but this adversity may also provide the impetus to alter a player's career pathway and lead to improvement. For example, "sometimes missing out drives that person to do something that they weren't doing originally" and "this is still such a subjective industry in that there will be people that will rate certain players."

The SOT, *current pathway* was largely reflective of the challenges associated with recruiting players from the Current AFL youth player pathway, including coaching for development rather than winning, the differences between state pathways and the multi-factorial process of talent identification. For example; "(for U18 coaches) is their big issue to win the premiership or to develop players to get drafted or is it to do both" and "more unstructured activities before they get to 15 years of age to allow them to work through it in their way." The SOT, *future* relates to the advancements within the industry and how the recruiters can improve their practice. Therefore, evolving game trends and rule changes, coupled with advancements in technology and the increase of data available are believed to have major impacts on the future working direction of the recruiter. For example, "we've just employed a full-time analyst to assist with the data analysis because we find it's hard to teach football people analytics" and

"There have been a few rule changes in previous years that have had some effect on what we do."

## Discussion

Talent identification in sport is a multifaceted, non-linear, dynamic process [18] due to the numerous factors associated with effective in-game performance (i.e., physical; physiological; technical; tactical; and psychological attributes) [8, 10, 15, 21]. While researchers recommend recruiters consider a holistic, multidisciplinary approach to talent identification [2, 17, 20], there is a limited understanding of elite sports recruiters talent identification decision-making processes. In the current study, we found the recruiting role to be relatively complex and multi-faceted by identifying 4 FOTs (i.e., The Recruiter; Processes and Practice; Assessment; and Selection), 25 SOTs and 179 TOTs. The research provides a detailed description of the role and associated influences on talent identification approaches within AFL recruitment.

### The recruiter

The findings from the current study highlight the factors associated with being a Australian Football recruiter. These factors include having a deep passion and love of the sport, typically developed through playing the game. In relation to their perceptions of the day-to-day decision-making, in support of previous findings [27, 28, 36], Australian football recruiters believe one of the most important personal attributes is previous talent identification experience, as this provides the key knowledge to inform their philosophy of talent, what they value in potential players. This can be described as tacit knowledge, which can be abstract and unarticulated [37], is largely gained experientially from direct experience handling every day (on-the-job) problems and increases with experience [38]. In a similar manner to coaching, where tacit knowledge shapes coaches decision-making abilities [29, 38], the recruiters in the current study indicated they make talent identification decisions based on similar decisions experienced in the past, recalling previous outcomes to enable appropriate decisions. As the talent identification process is more of an implicit rather than explicit process, recruiters use previous experience as a key component in this decision-making process [29], by searching for qualities reflecting those observed in players they have identified in the past that have developed into elite players. Recruiters simultaneously process and weigh information, that based on previous experience was counterproductive in the development of previously recruited players. As such, recruiters' tacit knowledge and what they value is not universal, but formed through personal life and talent identification experiences within the talent identification setting in which they are working, when discriminating between potential and non-potential players [29]. Therefore, an important consideration for talent identification researchers is how to assist recruiters in the development of this tacit knowledge, to potentially reduce the likelihood of recruiters new to a role making incorrect or poor talent based decisions.

From an academic perspective, there is a lack of a universally agreed definition for sporting talent [3–4], although, the results of the current study demonstrates elite Australian Football recruiters do have a clear definition and philosophy of sporting talent. Talent is defined broadly as a "capacity for achievement or success" [39, p. 3] and dependent on the "presence or absence of inborn attributes variously labelled as talents" [40, p. 399]; which "places a child among the top 10% of his or her age peers" [41, p. 67]. However, highlighted by the recruiters in this study, but missing from the literature definitions, is the acknowledgement of talent being a skill, or series of skills performed repeatedly and consistently. Therefore, it may be appropriate to consider including this dimension to the sporting literature definitions, to ensure they reflect the applied practice of the key talent identification stakeholders.

## Practices and processes

Within the talent identificaiton literature, there is limited understanding of the nuances associated with talent identification and selection at an elite performance level. The current study highlights the practices and processes talent recruiters undertake when making talent selection decisions. A key finding form the current study is the decisions they make are informed by what the individual recruiter values in an athlete, but also this decision is relative to the strategy and needs of the club. This has been supported in the literature, with MacMahon and colleagues [27] indicating Australian football recruiters need to have a strong understanding of the teams needs when making informed talent selection decisions. This is shaped by the relationship with the head coach, and the ability of the recruiter to adjust their decision-making style for the specific team context to ensure the club is always progressing [27, 42].

Further, the results provide an explicit understanding of the practices recruiters undertake when identifying a potential athlete. In relation to the main jobs recruiters undertake, a general consensus was each potential player is monitored over a two year period, which includes watching live game approximaltely 20 times with follow up observation via video footage. In addition to watching games, recruiters code specific game actions and have regular conversations about the potential player's performance with colleagues. In addition to this, recruiters are constantly collecting player data and cross-referencing, following up on discrepancies, referencing checking and conducting internal mock drafts. This process ensures the fine-tuning and calibration of the talent identification system and practices, to inform the decision-making process.

## Assessment

Talent identification is a dynamic and complex process, concerned with making informed decisions from current levels of performance to predict the most promising individuals who have the best potential to become an elite senior athlete [1–4, 16, 18, 19]. Despite extensive research in the talent identification area, researchers tend not to discuss the subtleties associated with identifying talent, but rather investigate athlete attributes associated with selection and non-selection via objective measures, with limited understanding or consideration of the recruitment staff opinions or practices. While post-hoc analyses of isolated physical performance measures indicate differences between identified and non-identified Australian football players [10, 15, 43], the current data suggests recruiters consider a variety of interdependent attributes when assessing potential elite level athletes. Therefore, researchers need to consider the development of multi-dimensional assessments which provide a more representative design [44]. As such, researchers should consider developing assessments representative of match play conditions, including the technical and tactical skills, and the conditions, actions and perceptual stimuli present during the competitive environment [45–49]. While isolated physical capability testing may provide a method of monitoring physical capabilities, representative assessments may provide more relevant information for recruiters when making talent identification decisions.

In line with this method of assessment, the current results highlight the importance of an athlete's in-game performance to being identified as talented. While researchers and associations still investigate the relationships between selection and non-selection on isolated physical and performance tests [10, 15, 43], there has been a recent shift in Australian Football to understand key in-game performance indicators influence on selection [11, 13, 14], coupled with the devleopment of more dynamic skill based assessments [46, 47]. Findings have indicated that match performance outcomes, such as contested possessions and delivering the ball into the forward 50-metre area, differentiate players drafted and not-drafted [14]. This

supports the recruiter's suggestions that players identified as talented do possess superior in-game technical abilities and may highlight the need for more opportunities for recruiters to observe potential players competing in match conditions.

Recently, researchers have found recruitment staff have shifted their focus away from isolated exercises where players exhibit physical qualities, to focus more on intrinsic qualities, such as gameplay intelligence and attitude [29]. This is further supported by researchers who have suggested sport-specific decision-making skills are important for successful in-game performance [50, 51], with researchers highlighting skill based differences in other invasion sports such as soccer [25, 52]. A recent investigation in youth Australian Football players found talent identified youth players are more accurate at making correct game-based decisions when compared to non-identified youth players on a video-based decision-making task [15]. This finding supports the recruiter's comments that game intelligence or decision-making ability is a key component of the talent identification decision-making process. While specific laboratory-based decision-making tasks are not used by recruiters when assessing player abilities, the Woods and colleagues [15] findings may indicate recruiters can use this information when forming opinions on potential athletes.

Consideration of psychological attributes is a critical component of optimal performance in sport, with the recruiters in the current study indicating players are potentially ruled out of selection due to the perception they could not mentally cope with the pressures of elite sporting competition. In recent years, researches have more closely examined the role of psychological attributes and mental skills from a talent identification and development (TID) perspective [18, 29, 53–55]. Not only is more evidence becoming available on positive psychological characteristics, but there is a growing appreciation of what MacNamara and Collins [55] described as the "darker" side of the human psyche. In the current study, expert recruiters were mindful of both the positive attributes (i.e., strengths-based approach) and negative attributes (i.e., deficit-based approach). Furthermore, the recruiters discussed psychological assessments, psychometric testing and the involvement of sport psychologists as an integral component of the overall talent pathway and selection process. While not a specific aim of the current investigation, future studies may consider exploring the specific psychological assessments considered important for talent identification and how recruiters use this information to inform talent identification decisions.

## Selection

The findings from the current study highlight the intricacies that inform the actual selection decisions by recruiters at the elite level of Australian football. The results indicate the selection of athlete's is also influenced by factors external to player performance measures. One factor in particular was the geographical location of the athlete and the recruiters perspective of the impact of this on their development. The recruiters indicated potential advantage of selecting athletes who had come from more remote locations due the perception these athletes would have a greater potential for improvement compared to athletes from more densely populated locations. This perception is supported by the sporting expertise literature, where researchers have found athletes born in smaller towns of cities (i.e., populations greater than 1000, but smaller than 500,000) were significantly over-represented in elite sports (i.e., baseball; basketball; golf; rugby league), compared an under-representation of athletes from cities greater than 500,000 residents [56, 57]. From a research perspective, the benefit of a smaller community during an athletes development may provide the athletes with a more facilitative environment for athlete development [56]. Therefore, as indicated by the recruiters, athletes form these communities may not have the opportunity to undertake structured athlete development

programs, but have potentially engaged in sporting endeavours from a unstructured perspective outlined by the Deliberate Play models of athlete development [58].

A key component of the recruiters selection practice is the constant reflection on previous talent identification decision-making situations. This includes personal experiences, and also reflecting on the AFL recruitment communities talent selection practices. In particular, there is the specific process of making talent predictions based on comparing current youth athletes directly to how a current elite level player was performing at the same age. This reflective process may demonstrate the recruiters using reflective practice processes, whereby an individual critically examines their experience with the intention of developing current knowledge and improving future performance [59]. Potentially due to this process, the recruiters are able to learn from personal effective, or ineffective talent selection decisions and more general league wide talent selection abnormalities to inform future talent selection decisions. While it was not a specific aim of this study, future investigations may look to build on this concept to determine how recruiters use reflection to guide their talent selection decision-making processes.

## Limitations

While this is one of the first studies to understand the role of the recruiter within the talent identification process, especially from their perspective, the findings should be considered with respect to several limitations. The findings provide the opinions and experiences of full-time Australian Football National Recruitment managers. While the participants are responsible for the final decision when selecting youth Australian Football players for their respective professional team, their opinions and experiences may differ to that of recruiters or scouts from different levels of experience and employment. As a result, it may be possible that similar studies utilising a range of participants including, full-time, part-time and volunteer recruitment staff may provide a greater range of experiences and perceptions into the factors which may contribute to talent identification and selection. Further, the current study is limited by the reflective nature of the data collection which provides a reflective understanding of the practices and processes associated with making talent identification decisions. Further research is required to empirically understand what the specific concurrent thought processes are of recruiters during the talent identification process. By incorporating a concurrent verbal reporting protocol (i.e., think aloud) it may be possible to identify the main performance attributes recruiters consider important when identifying talented youth Australian Football players [60].

## Conclusion

This study sought to understand the dynamic process of talent identification in Australian football. The findings provide a detailed description of Australian Football recruiter's knowledge and practice associated with the assessment and selection of youth athletes. From a practical perspective, the findings demonstrate that recruiters consider a variety of interdependent attributes, such as technical, tactical, physiological, psychological, perceptual-cognitive and game related performance to make an informed talent identification decision. Further, these decisions are underpinned by previous recruitment decisions (either positive or negative), personal talent identification philosophy, and club philosophy and player needs. While the results improve our understanding of the talent identification decision-making process of elite Australian football recruiters, researchers still need to consider potential methods to improve this process, such as the development of more rigorous objective instruments or testing procedures, for all stakeholders. In doing so, this will potentially improve the clarity of talent

identification decisions and increase our knowledge of the underlying skills and attributes associated with elite talent identification.

## Supporting information

**S1 Appendix.**
(DOCX)

## Author Contributions

**Conceptualization:** Paul Larkin, Damian Farrow.

**Data curation:** Paul Larkin, Daryl Marchant, Amy Syder, Damian Farrow.

**Formal analysis:** Paul Larkin, Daryl Marchant, Amy Syder.

**Funding acquisition:** Paul Larkin, Damian Farrow.

**Investigation:** Paul Larkin, Damian Farrow.

**Methodology:** Paul Larkin, Daryl Marchant, Amy Syder.

**Project administration:** Paul Larkin, Damian Farrow.

**Resources:** Paul Larkin.

**Writing – original draft:** Paul Larkin, Daryl Marchant, Amy Syder, Damian Farrow.

**Writing – review & editing:** Paul Larkin, Daryl Marchant, Amy Syder, Damian Farrow.

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
