## [Decision Letter · Decision Letter 0]

15 Sep 2020

PONE-D-20-13154

An Eye for Talent: The Recruiters’ Role in the Australian Football Talent Pathway

PLOS ONE

Dear Dr. Larkin,

Thank you for submitting your manuscript to PLOS ONE. After careful consideration, we feel that it has merit but does not fully meet PLOS ONE’s publication criteria as it currently stands. Therefore, we invite you to submit a revised version of the manuscript that addresses the points raised during the review process.

We look forward to receiving your revised manuscript.

Kind regards,

Fabrizio Perroni

Academic Editor

PLOS ONE

Journal Requirements:

2. Please include additional information regarding the survey or questionnaire used in the study and ensure that you have provided sufficient details that others could replicate the analyses. For instance, if you developed a survey guide as part of this study and it is not under a copyright more restrictive than CC-BY, please include a copy, in both the original language and English, as Supporting Information.

Reviewers' comments:

Reviewer's Responses to Questions

**Comments to the Author**

1. Is the manuscript technically sound, and do the data support the conclusions?

Reviewer #1: Yes

Reviewer #2: Yes

2. Has the statistical analysis been performed appropriately and rigorously? 

Reviewer #1: Yes

Reviewer #2: Yes

3. Have the authors made all data underlying the findings in their manuscript fully available?

Reviewer #1: Yes

Reviewer #2: Yes

4. Is the manuscript presented in an intelligible fashion and written in standard English?

Reviewer #1: Yes

Reviewer #2: Yes

5. Review Comments to the Author

Reviewer #1: This article highlighted the factors that shape talent identification processes in Australian Football league (AFL) recruiters. It is clear that recruiter’s decisions strongly influence the success of an AFL team, and from my understanding there was no peer-reviewed published information on how recruiters make their decisions on who they recruit. It is an interesting topic worthy of publication as it will fascinate the sporting community and budding athletes on the traits that are important to recruiters. The methodological approach was sound. Qualitative semi-structured interviews were conducted with 12 full time Australian Football recruiters. Thematic analysis resulted in the identification of four primary themes (the recruiter, processes and practices, assessment and selection). The researchers went into great depth to ensure the trust-worthiness of their data by documenting the process they implemented to ensure credibility, transferability, confirmability and dependability. The results were disseminated in first, second and third order themes which is a logical approach considering the diversity of recruiting philosophies, strategies, clubs requirements and recruiters backgrounds and training. The discussion was thorough, yet disciplined in focusing on relevant themes only and reinforced the results. I would have liked to seen a practical applications section outlining succinctly the most common themes that dictate how recruiters decides on who they recruit to provide the reader with easy to remember take-home messages as there was a lot of information to process from this research. Overall I found the document to be very well-written, logically structured with appropriate reference to relevant literature. This is a good foundation study that will stimulate further research into best practices for recruitment in AFL.

Page 26 Line 7: “addition to watching games, recruiters also have constant co and coding specific game actions”. What is constant co?

Reviewer #2: Thank you for the opportunity to review this article. I commend the authors for investigating a novel question of significance and using robust qualitative methodology. The article is well written and it is clear the authors have expertise in the area of research. Below are a few minor recommendations.

Introduction - Line 6 consider changing to "recruitment" decisions. Additionally, do the authors have a reference to support the statement about the ad nauseum scrutiny of the decisions.

In the methods section: The authors state that recruiters from 12 of the 18 clubs in the Australian Football League were interviewed. Are there multiple recruiters for each club? And what proportion of the total number of recruiters across the AFL were interviewed.

Page 8- lines 5-6: "7" should be spelled out in full

Page 17 line 13: Consider changing 'tactic' to 'tactical'.

General comment: There is a bit of overlap between the results and the discussion. Consider condensing the results section where appropriate and incorporating into the discussion. Additionally, discussing the limitations of the study within the discussion should be considered by the authors.

6. PLOS authors have the option to publish the peer review history of their article (what does this mean?). If published, this will include your full peer review and any attached files.

Reviewer #1: No

Reviewer #2: No

---

## [Author Response · Author response to Decision Letter 0]

18 Sep 2020

Reviewer 1

REVIEWER SUGGESTION 1: Overall I found the document to be very well-written, logically structured with appropriate reference to relevant literature. This is a good foundation study that will stimulate further research into best practices for recruitment in AFL. 

RESPONSE – We thank the reviewer for their time and the positive feedback about the manuscript. 

REVIEWER SUGGESTION 2: The discussion was thorough, yet disciplined in focusing on relevant themes only and reinforced the results. I would have liked to seen a practical applications section outlining succinctly the most common themes that dictate how recruiters decides on who they recruit to provide the reader with easy to remember take-home messages as there was a lot of information to process from this research

RESPONSE – We thank the reviewer for the positive feedback. While we are mindful of Reviewer 2’s comment in relation to the overlapping of information in the results and discussion, we have integrated the practical take-home message within the conclusion. This now states: “From a practical perspective, the findings demonstrate that recruiters consider a variety of interdependent attributes, such as technical, tactical, physiological, psychological, perceptual-cognitive and game related performance to make an informed talent identification decision. Further, these decisions are underpinned by previous recruitment decisions (either positive or negative), personal talent identification philosophy, and club philosophy and player needs.” (Page 23; Line 22 – Page 24; Line 2)

REVIEWER SUGGESTION 3: Page 26 Line 7: “addition to watching games, recruiters also have constant co and coding specific game actions”. What is constant co?

RESPONSE – We thank the reviewer for identifying this incomplete sentence within the manuscript. The sentence has not been revised to read: “In addition to watching games, recruiters code specific game actions and have regular conversations about the potential player’s performance with colleagues.” (Page 19; Lines 7 – 8) 

Reviewer 2

REVIEWER SUGGESTION 1: I commend the authors for investigating a novel question of significance and using robust qualitative methodology. The article is well written and it is clear the authors have expertise in the area of research. 

RESPONSE – We thank the reviewer for the positive comments about the manuscript.

REVIEWER SUGGESTION 2: Introduction - Line 6 consider changing to "recruitment" decisions. Additionally, do the authors have a reference to support the statement about the ad nauseum scrutiny of the decisions?

RESPONSE – We thank the reviewer for identifying this typographical error and have amended this in text. Further, the statement refers to anecdotal evidence from newspapers and social media, rather than empirical evidence, as such, we have modified the sentence to state “Anecdotally, recruitment decisions are retrospectively scrutinised ad nauseam by sports fans, the media and within sports organisations” (Page 2; Line 6). 

REVIEWER SUGGESTION 3: In the methods section: The authors state that recruiters from 12 of the 18 clubs in the Australian Football League were interviewed. Are there multiple recruiters for each club? And what proportion of the total number of recruiters across the AFL were interviewed.

RESPONSE – We thank the reviewer for the comment, and the need to clarify participant’s club associations. To do this, we have now highlighted that there were 12 participants in total, from 12 different club. The sentence now reads: “Twelve full-time male recruiters (i.e., scouts) from 12 of the 18 AFL professional clubs participated in this study” (Page 4; Line 24). Within the AFL talent identification system, there are varying levels of employment for recruiters, from volunteer to full-time. As such, we believed it was important to consider the individuals responsible for making the final talent recruitment decision, the National Recruitment Manager. This individual is full-time at the club and responsible for making the decision for each player who is recruited to the team. In addition to interviewing 66% of the leagues National Recruitment managers, by the 12th participant we were reaching data saturation. 

REVIEWER SUGGESTION 4: Page 8- lines 5-6: "7" should be spelled out in full

RESPONSE – We have amended this in text (Page 8; Line 5).

REVIEWER SUGGESTION 5: Page 17 line 13: Consider changing 'tactic' to 'tactical'.

RESPONSE – The word in the sentence highlighted by the reviewer is actually tacit, relating to tacit knowledge, as such no change has been made in the document. If we however have missed the word the reviewer is highlighting then we would be happy to change.

REVIEWER SUGGESTION 6: There is a bit of overlap between the results and the discussion. Consider condensing the results section where appropriate and incorporating into the discussion. Additionally, discussing the limitations of the study within the discussion should be considered by the authors.

RESPONSE – We thank the reviewer for the comment and acknowledge the potential overlap of information between the results and discussion. We have attempted to reduce some of the information within the results section where possible. However, due to the qualitative nature of the data, we do believe some detail is required to provide context around the themes and the associated quotes in the results section, with the discussion providing the links to the research around the identified themes. 

We thank to reviewer for identifying the lack of limitations within the document, and this was an oversight on our behalf. We have now included a limitations section in the manuscript (Page 23; Lines 1 – 18).

---

## [Decision Letter · Decision Letter 1]

13 Oct 2020

An Eye for Talent: The Recruiters’ Role in the Australian Football Talent Pathway

PONE-D-20-13154R1

Dear Dr. Larkin,

We’re pleased to inform you that your manuscript has been judged scientifically suitable for publication and will be formally accepted for publication once it meets all outstanding technical requirements.

Kind regards,

Fabrizio Perroni

Academic Editor

PLOS ONE

Additional Editor Comments (optional):

Reviewers' comments:

Reviewer's Responses to Questions

**Comments to the Author**

1. If the authors have adequately addressed your comments raised in a previous round of review and you feel that this manuscript is now acceptable for publication, you may indicate that here to bypass the “Comments to the Author” section, enter your conflict of interest statement in the “Confidential to Editor” section, and submit your "Accept" recommendation.

Reviewer #1: All comments have been addressed

Reviewer #2: All comments have been addressed

2. Is the manuscript technically sound, and do the data support the conclusions?

Reviewer #1: Yes

Reviewer #2: Yes

3. Has the statistical analysis been performed appropriately and rigorously? 

Reviewer #1: Yes

Reviewer #2: Yes

4. Have the authors made all data underlying the findings in their manuscript fully available?

Reviewer #1: No

Reviewer #2: Yes

5. Is the manuscript presented in an intelligible fashion and written in standard English?

Reviewer #1: Yes

Reviewer #2: Yes

6. Review Comments to the Author

Reviewer #1: (No Response)

Reviewer #2: I commend the authorship team for thoroughly addressing the comments on the original submission. The manuscript is much improved and is now at an acceptable standard.

7. PLOS authors have the option to publish the peer review history of their article (what does this mean?). If published, this will include your full peer review and any attached files.

Reviewer #1: No

Reviewer #2: No

---

## [Editor Report · Acceptance letter]

22 Oct 2020

PONE-D-20-13154R1 

An Eye for Talent: The Recruiters’ Role in the Australian Football Talent Pathway 

Dear Dr. Larkin:

I'm pleased to inform you that your manuscript has been deemed suitable for publication in PLOS ONE. Congratulations! Your manuscript is now with our production department. 

Kind regards, 

on behalf of

Dr. Fabrizio Perroni 

Academic Editor

PLOS ONE